# Enhancing polytetrafluoroethylene (PTFE) coated film for food processing: Unveiling surface transformations through oxygenated plasma treatment and parameter optimization using response surface methodology

Noraziani Zainal Abidin[1], Haslaniza Hashim[1]\*, Saiful Irwan Zubairi[1]\*, Mohamad Yusof Maskat[1], Noorain Purhanudin[2], Rozidawati Awang[2], Jarinah Mohd Ali[3], Harisun Yaakob[4]

**1** Department of Food Sciences, Faculty of Science and Technology, Universiti Kebangsaan Malaysia, UKM Bangi, Selangor, Malaysia, **2** Department of Applied Physics, Faculty of Science and Technology, Universiti Kebangsaan Malaysia, UKM Bangi, Selangor, Malaysia, **3** Department of Chemical and Process Engineering, Faculty of Engineering and Built Environment, Universiti Kebangsaan Malaysia, UKM Bangi, Selangor, Malaysia, **4** Institute Bioproduct Development (IBD), Universiti Teknologi Malaysia, Skudai, Johor, Malaysia

\* saiful-z@ukm.edu.my (SIZ); haslaniza@ukm.edu.my (HH)

## Abstract

Spray drying fruit juice powders poses challenges because sugars and organic acids with low molecular weight and a low glass transition temperature inherently cause stickiness. This study employed a hydrophobic polytetrafluoroethylene (PTFE) film to mimic the surface of the drying chamber wall. The Central Composite Design (CCD) using response surface methodology investigated the impact of power ($X_1$, Watt) and the duration of oxygenated plasma treatment ($X_2$, minutes) on substrate contact angle (˚), reflecting surface hydrophobicity. To validate the approach, *Morinda citrofolia* (MC) juice, augmented with maltodextrins as drying agents, underwent spray drying on the improved PTFE-coated surface. The spray drying process for MC juice was performed at inlet air temperatures of 120, 140, and 160˚C, along with Noni juice-to-maltodextrin solids ratios of 4.00, 1.00, and 0.25. The PTFE-coated borosilicate substrate, prepared at a radio frequency (RF) power of 90W for 15 minutes of treatment time, exhibited a porous and spongy microstructure, correlating with superior contact angle performance (171˚) compared to untreated borosilicate glass. Optimization data indicated that the PTFE film attained an optimum contact angle of 146.0˚ with a specific combination of plasma RF operating power ($X_1 = 74$ W) and treatment duration ($X_2 = 10.0$ minutes). RAMAN spectroscopy indicated a structural analysis with an ID/IG ratio of 0.2, while Brunauer-Emmett-Teller (BET) surface area analysis suggested an average particle size of less than 100 nm for all coated films. The process significantly improved the powder's hygroscopicity, resistance to caking, and moisture content of maltodextrin-MC juice. Therefore, the discovery of this modification, which applies oxygen plasma treatment to PTFE-

**Data Availability Statement:** All relevant data supporting the findings of this study are within the manuscript itself.

**Funding:** The authors express their gratitude to Universiti Kebangsaan Malaysia (UKM) for the financial support extended to this project (GUP-2020-080; ST-2023-043). Appreciation is also extended to the Department of Food Sciences and the Innovative Centre for Confectionery Technology (MANIS) within the Faculty of Science and Technology for their provision of research facilities and guidance.

**Competing interests:** The authors have declared that no competing interests exist.

coated substrates, effectively enhances surface hydrophobicity, contact angle, porosity, roughness, and ultimately improves the efficacy and recovery of the spray drying process.

## 1. Introduction

Recently, there have been noteworthy shifts in customer expectations regarding food products. The drying of fruit and vegetable juices is employed to produce a secure and convenient form that can be reconstituted into a high-quality product retaining the same nutritional and functional characteristics as the original juice [1]. The rising incidence of conditions such as overweight/obesity, cancer, diabetes, and other diseases, coupled with shifts in lifestyle, has resulted in an increasing need for bioactive food ingredients and nutraceuticals. These include phenolic compounds, carotenoids, essential oils, essential fatty acids, minerals, and vitamins [2].

Spray drying is a widely adopted drying technique in both the food and pharmaceutical sectors. Industrial systems boast capacities ranging from a few dozen kilograms to 40 metric tons of evaporated water per hour, meticulously chosen to comply with the demands of specific processes and production lines [3]. The adhesiveness of powders rich in carbohydrates is linked to the physical structure of the carbohydrate components, frequently found in an amorphous glassy state. The drying process induces a marked transition in both viscosity and surface tension at a composition- and temperature-dependent critical water activity level [4].

While spray dryers demonstrate versatility across a range of goods, specific issues tend to recur during the spray drying process. While certain issues have been tackled, others still demand attention and further research. These include issues with highly thermosensitive materials, stickiness in sugar-rich substances, and difficulties related to achieving proper particle size and distribution [3]. The occurrence of wall deposition presents a notable challenge in spray dryers. Given that spray drying commonly serves as the concluding stage in a process, issues related to wall deposition can directly affect the ultimate quality of the end products [5]. The spray drying of materials with low glass transition temperatures ($T_g$) presents notable challenges, as these materials are prone to particle cohesion, caking, and increased deposition on dryer walls. This can lead to reduced product yield and potential degradation in quality [1]. By integrating numerous innovative methods, manufacturers can significantly mitigate issues associated with adhesive particles in spray drying, leading to improved product quality, increased efficiency, and reduced production costs [6].

In the material-based approach, the glass transition temperature ($T_g$) of fruit juice and honey is elevated by incorporating a carrier substance with a high $T_g$. The dietary substitution of fiber-rich, protein-containing whole foods with refined carbohydrate-laden products, such as carriers, has been associated with adverse health outcomes, including obesity, diabetes mellitus, and hypercholesterolemia [7]. Process-based method concentrate on modifying spray drying parameters to mitigate powder stickiness. This involves employing techniques like surface scraping of the drying chamber, cooling the spray dryer walls, lowering the outlet air temperature to maintain the product temperature below the sticky point in the final stages of the drying process, and utilizing dehumidified air [1].

Maltodextrin's encapsulation properties during spray-drying minimize undesirable changes like flavor volatilization, pigment degradation, and component segregation. The inclusion of maltodextrin has been demonstrated to positively impact powder recovery in spray-drying applications [8]. Nevertheless, dryer design considerations may involve incorporating

windows into the walls for visual inspection, and issues related to glass design may arise, making the exploration of a metal design worthwhile. Components may be manufactured using a blend of metal and other materials, and the application of low surface energy materials to walls can reduce the adhesion of products, minimizing sticking [9]. Researchers are in the process of developing novel maltodextrins with modified structures, such as cyclic dextrins, which feature distinct branching patterns. These variations may result in reduced hygroscopicity and stickiness when compared to conventional linear maltodextrins.

Polytetrafluoroethylene (PTFE) exists in a polymerized tetrafluoroethylene form. Fluoro-based polymer coatings exhibit excellent anti-stick properties, but the aggressive environments and extreme conditions can degrade the coating, potentially compromising its anti-stick properties and other functionalities. This could involve chemical reactions, physical stresses, or thermal extremes. Anti-stick components in the food service industry commonly employ composite polytetrafluoroethylene (PTFE) coatings to improve their resistance to wear and tear and maintain hygienic conditions [10]. The distinctive spiral spatial arrangement of PTFE macromolecules, linked to the existence of fluorine in the primary chain, provides significant thermal resistance. However, it also renders the polymer vulnerable to irradiation, leading to the removal of fluorine from the polymer [11].

The intermolecular forces between these surfaces and liquids are weak, resulting in minimal wetting and a near-spherical drop shape [12]. There are various methods available to enhance the adhesion of PTFE. In certain instances, mechanical techniques such as surface roughening through sanding or grit blasting can effectively improve adhesion for particular applications.

Plasma-enhanced chemical vapor deposition (PECVD) is distinguished as a versatile technique for plasma polymerization in thin film fabrication, owing to its simplicity and the capability to be performed at room temperature [13].

PECVD resolves practical challenges linked to the wet chemical synthesis of thin films [14].

Compared to conventional polymers, films produced through PECVD are amorphous, possess a highly cross-linked structure, are insoluble, and display a pinhole-free dense formation. This results in enhanced chemical and physical stability, as well as strong adhesion to polymer substrates. The properties and sensitivity of plasma-treated PTFE films to contact angle can be customized by selecting suitable precursor gases and adjusting the deposition parameters of PECVD. Importantly, it appears that plasma treatment alone does not enhance the adhesion properties of PTFE. Consequently, surface modification strategies involving a combination of plasma treatment and graft polymerization, rather than adhesives, have been developed [15]. These discoveries indicate the potential to modify the characteristics of PTFE film surfaces, transitioning from hydrophobic to super-hydrophobic (with a contact angle exceeding 150 degrees). Although plasma treatment presents numerous benefits for surface modification, it also comes with certain limitations and potential drawbacks. Plasma operates by subjecting the surface to high-energy particles. Excessive power or prolonged treatment durations can result in surface damage, causing etching, cracking, or weakening of the material. Moreover, there is a risk of uneven treatment, with some areas being over-treated while others receive insufficient modification [6]. Moreover, the high-energy oxygen ions in the plasma engage with the surface of PTFE, leading to the fragmentation of carbon-fluorine bonds and the subsequent removal of material. Consequently, this procedure induces surface roughening, thereby increasing the surface area. However, there has been insufficient investigation into the aging behavior of PTFE surfaces treated with oxygen plasma in previous literature [16].

Response Surface Methodology (RSM), a statistical-mathematical approach, is employed to forecast and enhance multivariate statistical modeling equations using multiple regression analyses and optimization inquiries [17]. Response Surface Methodology (RSM) aids experimenters and designers in efficiently pinpointing processes or systems by utilizing quantitative

data obtained from well-designed experiments [18]. The optimization of process parameters aims to achieve more efficient and cost-effective conditions [19]. In a specific investigation, the optimization process focused on maximizing exergy efficiency, improvement potential rate, and sustainability index, while minimizing energy utilization, energy utilization ratio, and sustainability index [17].

Therefore, the primary objective of this study was to identify the optimum parameter for plasma treatment that could potentially enhance the contact angle of plasma-treated PTFE, consequently reducing particulate adhesion on the spray dryer wall chamber by creating a superhydrophobic surface that minimizes stickiness. The coefficient of determination $R_2$ yielded high values at 0.9647 and 0.9747, indicating the good agreement of the proposed model with the experimental responses. Maintaining a sustained hydrophobic condition is essential for preserving the efficiency of spray drying recovery (yield) throughout the process, providing a cost-effective solution for the food manufacturing industry. The improvement in the hydrophobicity of the PTFE membrane may be attributed to a reduction in powder stickiness. Future research will investigate the actual performance of spray drying at a pilot scale after demonstrating its effectiveness as the optimum method for increasing productivity and production output. Thus, to investigate the effectiveness of hydrophobicity post-oxygenated PTFE substrate, the study examined the impact of maltodextrin type and concentration during the spray drying of Noni juice at different drying temperatures on the key powder properties.

## 2. Materials and methods

### 2.1 Preparation of hydrophobic substrate

Microscope slides (Borosilicate, 76 mm x 26 mm x 1 mm) were procured from Quasi-S Technology Sdn. Bhd and were utilized to simulate the chamber wall of a spray dryer (Buchi Mini Spray Dryer B-290, Büchi, Flawil, Switzerland) A commercially accessible polytetrafluoroethylene (PTFE) tape, produced by the chemical company Hyunwoo Chemical in Korea, was selected for surface modification using plasma treatment. The specimens were sliced into square-shaped samples measuring 2.0 cm × 2.0 cm with a thickness of 0.2 mm for plasma treatment [20]. They were subjected to a cleaning procedure using pure ethanol followed by pure isooctane (GR for analysis, Merck) before being introduced into the plasma chamber.

### 2.2 Plasma treatment by oxygen gas

The plasma polymerization of oxygen on the PTFE substrate was conducted in a cylindrical quartz tube chamber with a 7-inch diameter. The process involved the use of a stainless-steel plate connected to an RF generator operating at a frequency of 100 MHz.

The volumetric gas flow rate was adjusted to 30 cm$^3$/minute and controlled through a mass flow controller (MKS). Prior to deposition, the samples underwent air plasma treatment (30W, 150 mTorr) for 2 minutes to eliminate contaminants on the membrane and activate its surface [6]. The separation distance between the two electrodes was kept at 3 cm, and the plasma was generated within the space between them. The glass slides underwent treatment under various conditions, maintaining constant treatment duration and oxygen gas proportion. The subsequent assessment concentrated on examining how these conditions affected the physico-chemical properties of the glass slides.

### 2.3 Measurement of Water Contact Angle (WCA)

The water contact angles on the surfaces of borosilicate glass and stainless-steel substrates were determined using a goniometer (Rame-Hart, Inc., Model 200–00), following the procedure

outlined by Sari [21] with minor adjustments. Contact angle values were averaged across five distinct points on the substrate surface ($n = 5$) [22, 23]. All measurements were conducted at ambient temperature ($29 \pm 1°C$). Contact angles were initially assessed both before and after the plasma treatments to evaluate the treatment's impact. Subsequent measurements were taken multiple times after the treatment to gauge hydrophobic recovery. The monitoring of hydrophobic recovery occurred under two distinct conditions [24]. Contact angle analysis was performed using the goniometer (Rame-Hart, Inc., Model 200–00) over a span of 14 days.

## 2.4 Scanning Electron Microscopy (SEM) and Energy Dispersive Spectroscopy Analysis (EDX)

The samples were examined using a scanning electron microscope, specifically the JSM-5800 model made by JEOL, Japan, coupled with Energy Dispersive Spectroscopy Analysis (EDX). Each EDX spectrum was normalized by the counting rates (cts/s) of the coupled EDX detectors and plotted against the keV scale. This approach enables an expert to observe the characteristic morphology and examine the elemental content of the sample or object simultaneously, without causing any pre-existing damage. To enhance visibility, all samples underwent a process of sputtering in a vacuum to apply a fine layer of carbon. The procedure requires the operator to arrange the layout of the stubs and the standard, specify the expected chemical classes of the particles, establish the limits of particle size, and define both the lower and upper particle diameter within the analyzed area [25].

## 2.5 Designing experiments for surface coating using Central Composite Rotatable Design (CCRD) with Response Surface Methodology (RSM)

Regarding previous experiments, it was observed that PTFE-coated plates treated with oxygen plasma exhibit significantly higher hydrophobicity in comparison to borosilicate glass. For the optimization of plasma operation surface conditions, RSM was applied, utilizing the Central Composite Design (CCD) with two primary variables.: $X_1$ for plasma operating power (W) and $X_2$ for duration of time (min). The design was autonomously generated using Design-Expert software version 13.0. The adequacy of the models was assessed by conducting a significant one-way analysis of variance (ANOVA), an inconsequential lack-of-fit test, and ensuring a coefficient of determination ($R^2$) exceeding 0.75 [26]. Table 1 displays the real and coded

**Table 1. Real and coded values for the power (watt) of the plasma system ($X_1/x_1$) and the duration of time (minute) ($X_2/x_2$) in the optimization process for hydrophobic coating on borosilicate glass using PTFE films.**

| Run | $X_1$ ($x_1$) (power, W) | $X_2$ ($x_2$) (time, minute) |
|---|---|---|
| 1 | 3.43146 (-1.414) | 25 (0.000) |
| 2* | 60 (0.000) | 25 (0.000) |
| 3 | 116.569 (1.414) | 25 (0.000) |
| 4 | 60 (0.000) | 46.2132 (1.414) |
| 5* | 60 (0.000) | 25 (0.000) |
| 6* | 60 (0.000) | 25 (0.000) |
| 7* | 60 (0.000) | 25 (0.000) |
| 8 | 100 (1.0) | 40 (1.000) |
| 9 | 100 (1.0) | 10 (-1.000) |
| 10 | 20 (-1.000) | 40 (1.000) |
| 11 | 60 (0.000) | 3.7868 (-1.414) |
| 12 | 20 (-1.000) | 10 (-1.000) |
| 13* | 60 (0.000) | 25 (0.000) |

values representing the power (watt) of the plasma system ($X_1/x_1$) and the duration of time (minute) ($X_2/x_2$) involved in the optimization process for applying hydrophobic coating onto borosilicate glass using PTFE films. Following that, the chosen models were subjected to optimization, with the criteria geared towards achieving a superhydrophobic effect by minimizing the relevant values.

## 2.6 Raman spectroscopy analysis

Raman spectroscopy is a vibrational spectroscopic technique for the characterization of catalyst materials, providing information about the catalysts' (defect) structure in the bulk and at the surface as well as the presence of adsorbates and reaction intermediates [27]. Raman spectra were acquired using a Thermo Scientific DXR Raman system featuring a microscope, a motorized microscope stage sample holder, and a charge-coupled device (CCD) detector. The excitation laser had a wavelength of 532 nm. The spectral range covered from 3500 to 100 cm-1, with a grating resolution of 5 cm$^{-1}$. The data acquisition involved 8 scans with an exposure time of 46 s/scan.

## 2.7 BET surface area analysis

The specific surface area (SSA) is of considerable importance in the adsorption of chemicals and ions from aqueous or gaseous solutions, playing a pivotal role in nanoparticle applications [28, 29]. The specific surface area (SSA) is particularly noteworthy as the most crucial morphological characteristic for solid substances, especially in applications involving porous structures like industrial adsorbents, catalysts, pigments, cement, and polymers [30]. Micropores, mesopores, and macropores are categorized based on their diameters: $d_{pore}$ <2 nm for micropores, 2 nm to 50 nm for mesopores, and exceeding 50 nm for macropores [31]. The surface areas of the PTFE-coated substrate and borosilicate glass, subjected to oxygen plasma treatment, were quantified using a surface area analyzer gas absorption BET (ASAP® 2020; Micromeritics® Instrument Corporation, United States of America) through nitrogen adsorption isotherms at 77 K. Before conducting experiments, a "degassing" step was implemented to remove surface contaminants, and samples were vacuum-dried overnight (12 hours) at temperatures below their glass-transition (Tg) temperature (110°C) [32].

## 2.8. Efficacy process: Preparation of spray drying

Noni fruits (*Morinda citrofolia*) were diced into smaller pieces and then blended into a liquefied puree using a food blender (7011S, Waring Blender, Torrington, USA) at a ratio of 2:1 (w/v). Subsequently, the juice underwent filtration with muslin cloth and centrifugation at 9500 rpm using a laboratory homogenizer (Sorvall HS 23, Thermo Electron Corporation, USA) for 15 minutes. Centrifugation facilitated the removal of fruit pulp from the extract. The feed containing maltodextrin was subjected to spray-drying in a Büchi B-290 mini spray-dryer (Buchi Mini Spray Dryer, Flawil, Switzerland). The dryer operated at an air flow rate of 900 m$^3$/minute, maintaining constant aspirator and pump rates at 100 and 10%, respectively [8]. Spray drying of Noni juice occurred at inlet air temperatures of 120, 140, and 160°C, with Noni juice-to-maltodextrin solids ratios of 4.00, 1.00, and 0.25. Throughout the spray-drying process, the feed mixtures containing maltodextrin were continuously stirred to ensure homogeneity [33]. The resulting spray-dried Noni fruit powders were collected from the product vessel, vacuum packed, and stored in dark conditions at -20°C until further analysis.

## 2.9 Hygroscopicity, moisture content and degree of caking

The hygroscopicity of Noni powder was evaluated following the procedure outlined by Cai and Corke [34]. Two grams of the powder were accurately weighed into pre-weighed Petri dishes and placed in an airtight desiccator filled with a saturated solution of $Na_2SO_4$ (81% RH) at 25˚C for one week. Hygroscopicity was quantified as grams of adsorbed moisture per 100 grams of dry solids [8]. The percentage weight loss of the powder (% w/w), determined by oven-drying until a constant weight was achieved, was used to calculate the moisture content (%) of the stored powder [35]. After the hygroscopicity determination, the wet sample underwent drying in an oven at 70˚C. Subsequent to cooling, the dried sample was weighed and transferred to a sieve with a size of 500 mm. The sieve was then shaken for 5 minutes using a shaking apparatus. All analyses were performed in triplicate ($n = 3$).

## 2.10 Statistical analysis

The experimental design and statistical analysis were conducted using Design-Expert software version 13 (Stat Ease, Minnesota, MN, USA). The collected data underwent analysis of variance (ANOVA) and Duncan's tests using the Statistical Analytical System (SAS®) version 22 (SAS Institute Inc., Cary, NC, USA). All experiments were conducted in triplicate ($n = 3$).

# 3. Result and discussion

## 3.1 Water Contact Angle (WCA) degradability profiles

The contact angle profiles illustrated in Fig 1 provide evidence that the hydrophobicity of the PTFE membrane substrate can be enhanced through the immobilization of radicalized oxygen plasma on its surface. An increase in water contact angle (WCA) has been documented for PTFE substrates subjected to a 15-minute air plasma treatment. The WCA values for the plasma-treated PTFE reached 171˚, indicating superhydrophobicity, a noticeable improvement from the 118˚ observed in the untreated Teflon. Initially, PTFE exhibits a hydrophobic

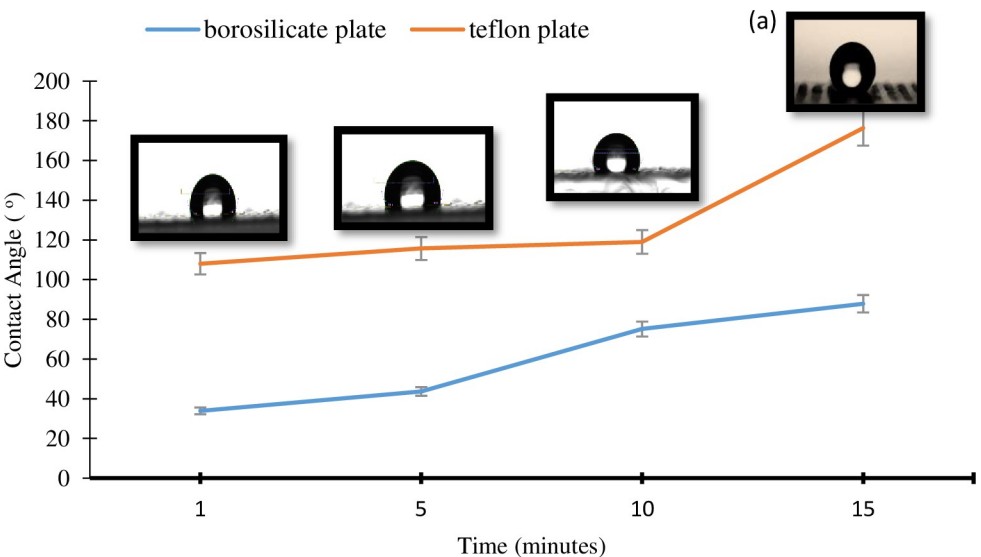

**Fig 1. Variations in the water contact angle of PTFE- and borosilicate-coated substrates exposed for different periods, ranging from 1 to 15 minutes.** The inset pictures (a-d) depict the dynamic changes as the exposure time progressively extends (15 minutes on top).

nature, with a static WCA of 111.4 ± 6.2˚ before isooctane cleaning and 103.6 ± 8.8˚ after cleaning [25], while reported values range from 102.5˚ to 130.8˚ [12].

This study successfully surpassed the hydrophobicity threshold, as confirmed by the FESEM images in Fig 2, which depicted physical and chemical changes. The static WCA increased to over 90˚ following exposure to radicalized $O_2$ plasma for 1, 5, 10, and 15 minutes under atmospheric pressure (Fig 1). Furthermore, high-power oxygen plasma treatment resulted in a significant WCA increase compared to a borosilicate glass substrate ($p<0.05$), reaching values exceeding 160˚ at a high radiofrequency (RF) power of 90W (Fig 1). Unlike PTFE, borosilicate glass displayed a distinct contact angle shift towards hydrophilicity after plasma treatment. Previous studies show that water droplets on hydrophobic surfaces act as tiny magnets, attracting and concentrating dissolved solutes [36] due to the condensation effect on hydrophobic surfaces. Plasma-induced polymerization of radicalized and oxidized $O_2$ molecules at high exposure power (90W, 15 minutes) in this study resulted in a dramatic increase in the WCA (171˚), creating a superhydrophobic surface.

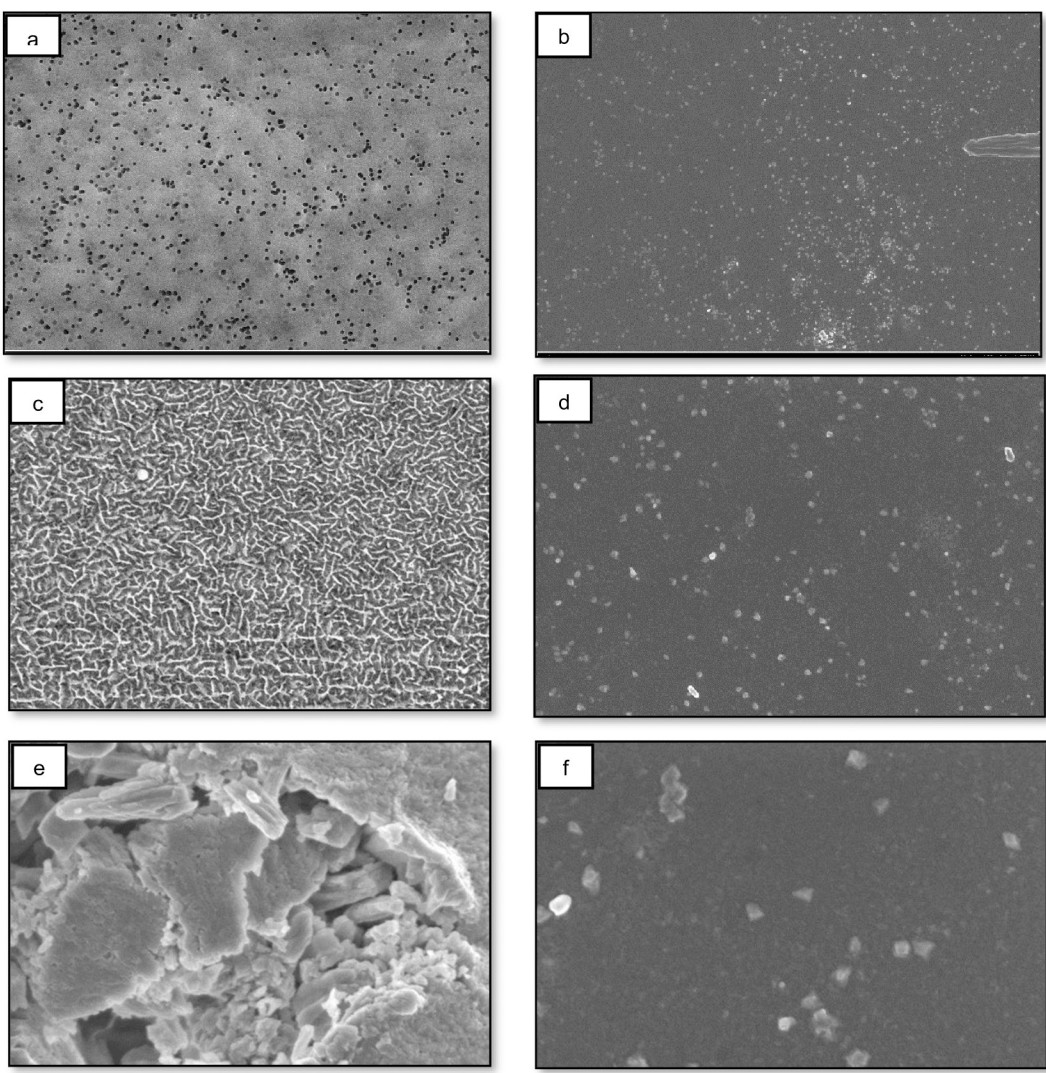

**Fig 2.** FESEM images of the PTFE-coated film substrate samples (a, c, e) and borosilicate samples (b, d, f) subjected to oxygen plasma treatment at 90W for 15 minutes, captured under magnifications of 5000x (a, b), 10000x (c, d), and 30000x (e, f) respectively.

**Table 2. Degradability profiles of water contact angle (WCA) for both borosilicate glass and PTFE-coated film substrates after oxygen plasma treatment, illustrated in relation to the treatment time.**

| Treatment exposure (minute) | *WCA Initial exposure (°) | *WCA after 15 days (°) |
|---|---|---|
| Untreated borosilicate glass | 48.20 ± 0.7 | 47.10 ± 1.1 |
| 1 | 43.30 ± 0.2 | 39.10 ± 0.8 |
| 5 | 40.60 ± 0.7 | 38.60 ± 0.1 |
| 10 | 38.10 ± 0.3 | 33.40 ± 0.8 |
| 15 | 30.20 ± 1.2 | 28.40 ± 0.8 |
| Untreated PTFE film | 118.10 ± 0.5 | 106.10 ± 0.3 |
| 1 | 108.80 ± 0.6 | 107.10 ± 0.2 |
| 5 | 116.30 ± 0.7 | 116.10 ± 0.5 |
| 10 | 118.90 ± 1.4 | 117.31 ± 0.8 |
| 15 | 171.00 ± 0.1 | 170.20 ± 0.3 |

*Contact angles (CA) were calculated from the average of six measurements taken for each time point ($n = 6$); The storage condition for hydrophobic recovery was set at room temperature (28 ± 1˚C; Relative humidity (RH): 75–80%)

Table 2 displays the degradability profiles of water contact angle (WCA) for surface samples of borosilicate glass and PTFE film following oxygen plasma treatment and a 15-day storage period. The treated samples were securely kept in an air-tight container filled with atmospheric air until surface characterization using water contact angle measurements. Hydrophobic recovery conditions were maintained at room temperature (28 ± 1˚C) with a relative humidity (RH) range of 75–80%. The gradual changes in WCA, decreasing from 118˚ (untreated PTFE) to 116.3˚ and 108.8˚ after 5 minutes and 1 minute of exposure, respectively, are likely due to the hydrophobic recovery process. This process can adjust the surface free energy, impacting material properties like wettability and adhesion. The recovery mechanism of hydrophobicity is influenced by various material- and process-dependent factors, including precursor selection, substrate characteristics, treatment parameters, and the micro-environment during aging, all intertwined with enthalpy considerations [22]. In fact, liquid immersion notably accelerates hydrophobic recovery timescales in LDPE films compared to dry environments like vacuum or humid air [37]. Water Contact Angle (WCA) underscores the significance of precise measurement and control of surface properties to optimize performance and facilitate innovative applications, including mechanisms for altering surface hydrophobicity. Hence, specific temperature ranges could potentially play a crucial role in the effective operation of the hydrophobic recovery mechanism.

## 3.2 Surface characteristics of borosilicate glass and PTFE-coated

The treated PTFE-glass surface (a) exhibited a substantial increase in surface roughness compared to the untreated glass surface (b). Adsorption of radicalized oxygen ions onto reoriented PTFE functional groups could be responsible for the observed increase in surface roughness, resulting in a decrease in wettability. Fig 2 presents SEM images illustrating PTFE and non-PTFE samples subjected to PEVCD treatments of varying durations. Systematic variations in plasma treatment parameters, particularly treatment time, lead to corresponding alterations in the roughness and dimensions of features on PTFE surfaces.

Additionally, extended treatment durations led to a significant rise in the contact angle value, attributed to the surface becoming more textured. Prior research has indicated a marked reduction in molecular adhesion as the surface contact angle increases, linked to the enhanced

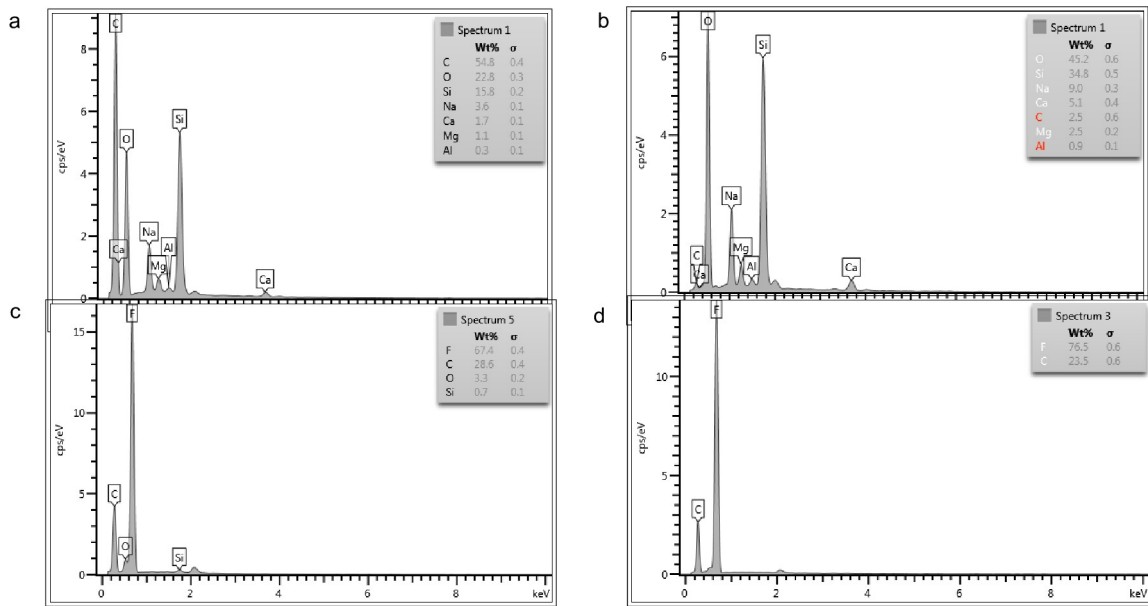

**Fig 3.** Labelling of elemental composition in EDX spectra for borosilicate glass samples before oxygen plasma treatment (a, b) and PTFE film substrates after plasma treatment (c, d) at a treatment duration of 15 minutes and 90W (power).

roughness of the glass surface resulting from plasma treatment [38]. In the quest to develop superhydrophobic surfaces, different techniques have been investigated, and one noteworthy example is the plasma-based dry etching method that utilizes reactive gases, as demonstrated in a previous study. It is important to highlight that various polymer experience etching at different rates, resulting in the creation of a uneven surface in composite etching situations [39].

Fig 3 illustrates the variances in elemental species observed before and after oxygen plasma treatment via EDX. Within the realm of oxygen plasma treatment, two conflicting reactions occur: the initial one involves surface modification, leading to fluorine depletion and the introduction of oxygen; the second one entails etching, producing a surface that is chemically like PTFE but exhibits a spongy-like morphology (Fig 2(F)). Initially, surface modification takes precedence but is eventually eclipsed by the process of etching. According to the EDX results, the augmentation of fluorine element content in PTFE corresponds to heightened hydrophobicity, as demonstrated by increased contact angles [40]. A stable state is achieved, presenting a surface that is no longer chemically akin to PTFE but comprises more components containing fluorine than at the beginning of the treatment. Extended exposure results in a dynamic evolution of micro-topography.

The existence of non-polar functional groups on typical glass surfaces hinders the penetration of water molecules into the prevailing polar bonds, consequently improving the hydrophobic nature [41]. The EDX profiles in Fig 3 reveal that PTFE surfaces initially contained up to 3.3% oxygen before oxygen plasma treatment, with this concentration diminishing to zero upon exposure. In contrast, borosilicate glass surfaces exhibited an increment in oxygen content (wt%) from 22.8 to 34.8 before oxygen plasma treatment. The chemical interaction of reactive oxygen species with the borosilicate glass surface induces spatial oxidation, reducing the water contact angle (WCA) and resulting in surface cleaning. Discrepancies in WCA compared to other studies may be attributed to differences in plasma conditions. For instance, prior work [42] employed only 3W of plasma power, whereas the present study utilized 90W with varying treatment durations.

### 3.3 Optimised responses of oxygenated plasma treatment PTFE film surface

The equation for the response surface, obtained by fitting contact angle data according to the models, is presented by Eq (2) and Eq (3). Through variance analysis, both models were deemed significant, with $R^2$ values exceeding 0.75, indicating a robust fit. Additionally, the lack-of-fit test for both substrates was inconsequential, affirming a strong alignment between experimental data and the model.

Two factorial interaction (2FI) models were established to elucidate the impact of independent deposition process variables on contact angle. Analyzing the experimental data generated by the software's central composite design (CCD), ANOVA analysis unveiled a notably high F-value of 41.23 for the selected model, signifying its significance. Furthermore, the *p*-value, representing the probability of error, was less than 0.05 (<0.0001), indicating that both power and duration time exert a significant effect on the contact angle value at a 95% confidence level. Table 4 displays the ANOVA results for the 2FI model.

The Model F-value of 41.23 indicates the model's significance. The *p*-values below 0.0500 suggest that model terms hold significance, while values above 0.1000 imply the opposite. The lack-of-fit F-value of 0.63 suggests that the lack-of-fit is not significant compared to pure error. A non-significant lack-of-fit is favorable, indicating the model is suitable and fits well. The predicted $R^2$ of 0.8440 aligns reasonably with the adjusted $R^2$ of 0.9096, with a difference less than 0.2. This negligible lack-of-fit value further supports the model's appropriateness in describing the response. Furthermore, the calculated $R^2$ value of the model closely approaches 1 and exhibits minimal deviation from the adjusted $R^2$ value. This consistency strengthens the appropriateness of the generated model in describing the observed response.

To enhance the contact angle value of the surface plate, optimization of the plasma system's power (watt) and duration time (minutes) was conducted employing a 2FI model within the experimental range of independent variables (Table 3). The root-mean-squared deviation (RMSD) value, which gauges the difference between expected and experimental data, was acquired through repeated tests to evaluate the model's validity. The validation of the optimal

**Table 3. The observed values for the response variable, the contact angle (CA), and the actual levels of the independent variables for the PTFE-coated film plate.**

| Run | A: Power | B: Duration time | Contact angle |
|---|---|---|---|
| | Watt | Minute | Degree |
| 1 | 3.43146 | 25 | 119.5 |
| 2* | 60 | 25 | 145.9 |
| 3 | 116.569 | 25 | 168.4 |
| 4 | 60 | 46.2132 | 160.9 |
| 5* | 60 | 25 | 156.6 |
| 6* | 60 | 25 | 151.4 |
| 7* | 60 | 25 | 149.1 |
| 8 | 100 | 40 | 173.1 |
| 9 | 100 | 10 | 162.9 |
| 10 | 20 | 40 | 153.8 |
| 11 | 60 | 3.7868 | 132.3 |
| 12 | 20 | 10 | 118.4 |
| 13* | 60 | 25 | 141.2 |

*Replication of centre point

point was ascertained using root-mean-squared deviation (RMSD), as detailed and represented by Eq (1) [43] below:

$$RMSD = \sqrt{\frac{1}{n-1}\sum_{i=1}(\hat{y}_i - y_i)^2} \tag{1}$$

$\hat{y}_i$ = experimental value;
$y_i$ = predicted value;
$n$ = number of samples.

The equation, expressed in terms of coded factors, enables predictions about the response for specific levels of each factor. Typically, high levels of factors are coded as +1, while low levels are coded as -1 by default. This coded equation proves valuable for discerning the relative impact of factors through a comparison of their coefficients. Following the above analysis, the models proposed for estimating the contact angle, generated using Design Expert software, adopted 2FI (two-factor interaction) models. Eqs (2) and (3) illustrate the empirical interactions between the actual and coded equations, respectively:

$$Y_1 = 90.12534 + 0.677985\ x_1 + 1.34705\ x_2 - 0.010500\ x_1 x_2 \tag{2}$$

$$Y_2 = 148.73 + 16.62x1 + 10.76x_2 - 6.30x_1 x_1 \tag{3}$$

## 3.4 Validation of optimized responses

To validate the optimization process, an experimental laboratory test was conducted at the selected optimum condition recommended by the RSM (Table 4) and Eqs (2) and (3). Table 5 demonstrates that the predicted values for all responses closely align with the actual values measured in the lab. The high accuracy value (close to 100%) and low error and standard deviation values between the predicted and actual values confirm the effectiveness of the optimization process. This suggests that the models accurately capture the relationship between the factors and responses.

Table 6 presents coefficient estimates, indicating the expected change in response per unit change in factor value when all other factors are held constant. In an orthogonal design, the intercept represents the overall average response of all the runs. Adequate Precision, measuring the signal-to-noise ratio, is desirable at a ratio greater than 4. The ratio of 19.975 indicates an adequate signal, allowing this model to navigate the design space. Subsequent repetitive tests were carried out using the optimal parameter values. According to Table 7, the predicted contact angle value was 146.0˚ with a desirability value of 0.851. The RMSD value, calculated using Eq (1), indicates a small RMSD value (3.4), confirming the suitability of the chosen model.

**Table 4. ANOVA results for the contact angle obtained from the Response Surface 2FI Model.**

| Source | Sum of squares | df | Mean square | F-value | p-value | Remarks |
|---|---|---|---|---|---|---|
| **Model** | 3293.89 | 3 | 1097.96 | 41.23 | < 0.0001 | significant |
| **A-Power** | 2209.63 | 1 | 2209.63 | 82.97 | < 0.0001 | |
| **B-Duration of time** | 925.50 | 1 | 925.50 | 34.75 | 0.0002 | |
| **Residual** | 239.68 | 9 | 26.63 | | | |
| **Lack of fit** | 105.83 | 5 | 21.17 | 0.6325 | 0.6896 | not significant |
| **Pure error** | 133.85 | 4 | 33.46 | | | |
| **Cor total** | 3533.57 | 12 | | | | |

**Table 5. Comparing the anticipated contact angle values of the PTFE-coated substrate with the actual contact angles obtained in repeated trials using optimum parameters to assess the validity of the model.**

| Response | Optimum power (W) | Optimum duration time (minute) | Predicted contact angle value (˚) | Contact angle value from the actual experiment | RMSD |
|---|---|---|---|---|---|
| Contact angle | 74 | 10 | 146˚ | Trial 1: 151.1˚<br>Trial 2: 143.8˚<br>Trial 3: 145.9 | 3.4 |

Figs 4 and 5 depict the normal plot of residuals and a three-dimensional response surface, respectively. The normal plot of residuals indicates a well-fitted normal percent probability, with responses aligning with straight lines that suggest implications for the design space [44]. The model's suitability to predict the highest contact angle at various durations of time and power was further assessed using both predicted vs. actual and 3D response surface plots in Figs 4 and 5, respectively. Generally, points significantly above or below the diagonal line may be considered over- or underestimated by the model [45].

### 3.5 Raman spectroscopy profiles

The Raman spectra of both the PTFE-coated substrate and the borosilicate glass treated under oxygen plasma are shown in Fig 6(A) and 6(B). The sample spectra exhibit two bands at 1378.95 and 1593.88 cm$^{-1}$, assigned to the G and D bands, respectively. The ratio of the D to G band intensities (ID/IG), serving as a measure of disorder in the carbon lattice, is 0.25. In contrast, the borosilicate glass substrate does not possess an ID/IG value. Additionally, the ID/IG values measured and presented in Table 6 can be utilized to assess the size of the sp2-domains La, where λL is the wavelength (in nm) of the exciting laser. These domains, identified as the most intense peaks of PTFE in its Raman spectra (at 289 and 732 cm$^{-1}$), were noted in a previous study [46].

The peak intensity of the borosilicate glass substrate in Fig 6 is notably heightened at 560.93 cm$^{-1}$ and 1095.69 cm$^{-1}$. In the analysis of Raman spectroscopic data for borosilicate glasses, three regions can be arbitrarily distinguished: 300–800, 800–1200, and 1200–1700 cm$^{-1}$ [47]. Additionally, the spectrum features broad bands in the 900–1200 cm$^{-1}$ region, ascribed to vibrations of an uncrystallized silicate component. Based on the above mentioned finding, the Raman spectroscopy serves as a valuable tool for authenticating food products by identifying distinctive molecular fingerprints linked to specific ingredients or characteristics. Food manufacturers can maintain product quality and safety by monitoring these surface alterations and adjusting processing and storage conditions accordingly.

### 3.6 BET (Brunauer–Emmett–Teller) surface area profiles

Fig 7 illustrates the sorption isotherms for the PTFE-coated substrate at optimum parameters, comparing it with borosilicate glass. For certain materials, distinguishing between surface adsorption and micropore filling can be challenging, though pore size distributions can be estimated using the Brunauer-Emmett-Teller (BET) equation [48]. As per Table 7, the BET surface area for the PTFE-coated substrate is 0.129 m$^2$/g, with an average pore size of 55.36 nm.

**Table 6. The positions of primary bands, full widths at half-maximum (ΓX), and intensity ratios of the bands in the Raman spectra.**

| Sample | Band (D), cm$^{-1}$ | ΓD, cm$^{-1}$ | Band (G), cm$^{-1}$ | ΓG, cm$^{-1}$ | I$_D$/I$_G$ | L$_a$, nm |
|---|---|---|---|---|---|---|
| Glass | N/A | N/A | N/A | | | |
| PTFE with oxygen plasma treatment at optimum parameter | 1378.95 | 40.6 | 1593.88 | 20.4 | 0.25 | 86 |

**Table 7. BET surface area comparison between the PTFE-coated film substrate and the borosilicate glass substrate.**

| Sample | Glass | PTFE |
|---|---|---|
| BET Surface Area ($m^2/g$) | 0.07730 | 0.1289 |
| Pore volume ($cm^3/g$) | 0.000053 | 0.000715 |
| Average pore size (nm) | 4.5612 | 55.3596 |

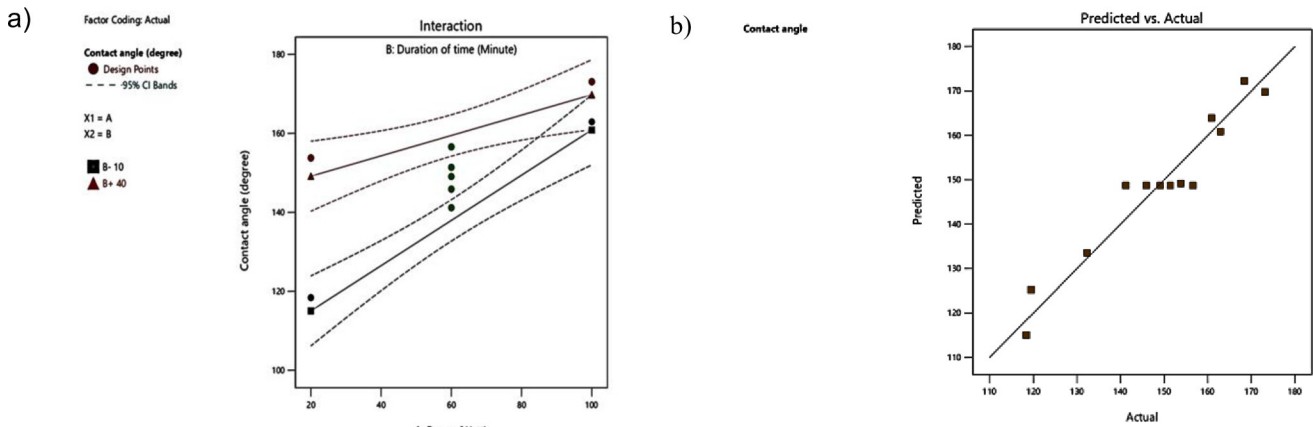

**Fig 4.** (a) The real interaction depicted in the Design-Expert plot; (b) The Design-Expert plot comparing predicted vs. actual results.

**Fig 5. 3D response surface plot generated by Design Expert, illustrating the relationship between contact angle, power (watt) of the plasma system, and duration of time (minutes).**

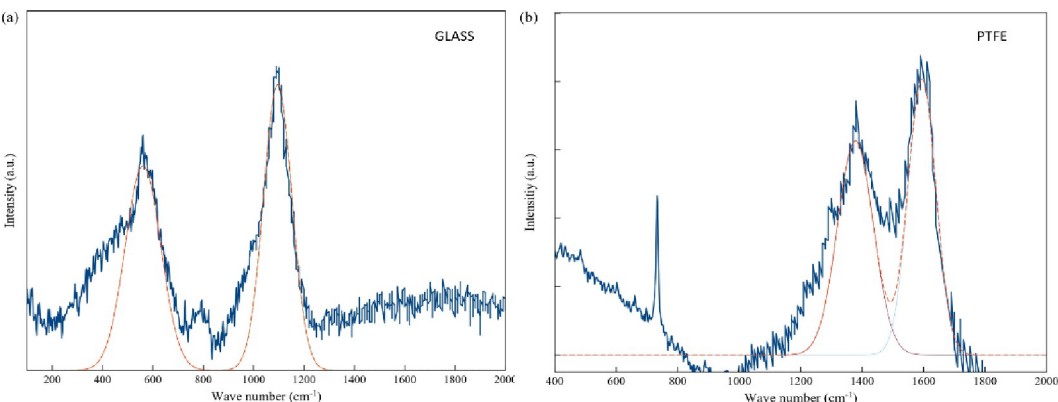

**Fig 6. Raman spectra obtained from the PTFE-coated film substrate following oxygen plasma treatment.**

During low-temperature plasma treatment, $O_2$ molecules are known to penetrate and react within the interior of PTFE, leading to the opening of internal porosity and a gradual increase in surface area [49]. The BET surface area of borosilicate glass is documented as 0.077 m2/g, accompanied by an average pore size of 4.561 nm. In physisorption, the dominant factor is the energetic interaction between guest molecules and surface atoms, surpassing interactions among the guest molecules themselves. Therefore, it is anticipated that a higher surface area per unit mass of material will correspond to a greater quantity of adsorbed material per unit

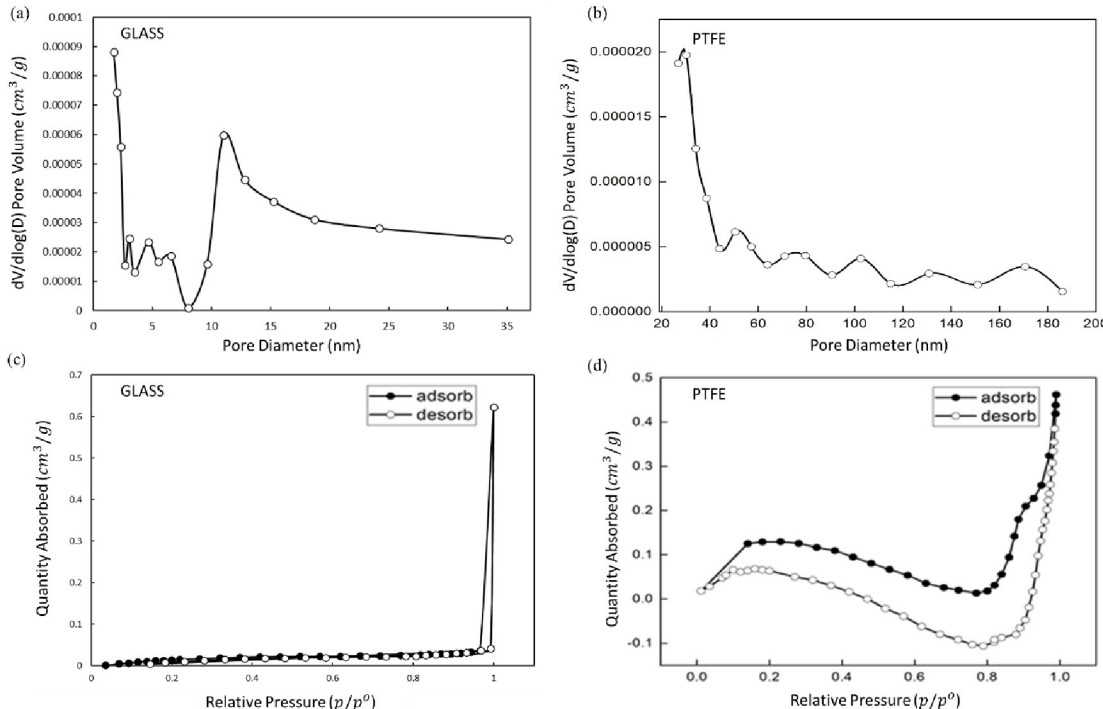

**Fig 7.** (a) Pore volume plotted against pore diameter for the borosilicate glass substrate; (b) Pore volume against pore diameter for the PTFE-coated substrate. (c) Sorption isotherm plot against pore diameter on the borosilicate glass substrate; (d) Sorption isotherm plot against pore diameter on the PTFE-coated film substrate.

**Table 8.  Impact of varied drying conditions (inlet/outlet temperature) and maltodextrin concentration on the physical properties of Noni powders using a PTFE substrate treated with post-oxygenated plasma as the drying surface.**

| Inlet Temperature (˚C) | Noni juice-to-maltodextrin solids ratio at 25˚C | Moisture content (%) | Hygroscopicity (g/100 g) | Degree of caking (%) |
|---|---|---|---|---|
| 120 | 4 | 2.65 ± 0.11[ap] | 11.90 ± 0.09[cq] | 3.54 ± 0.02[cr] |
|  | 1 | 4.14 ±0.03[aq] | 10.43 ± 0.11[bp] | 5.35 ± 0.1[cp] |
|  | 0.25 | 4.87 ± 0.05[aq] | 10.67 ± 0.05[br] | 5.62 ± 0.05[bp] |
| 140 | 4 | 2.38 ± 0.08[ap] | 13.17 ± 0.04[br] | 3.14 ± 0.04[cp] |
|  | 1 | 3.12 ± 0.05[aq] | 12.05 ± 0.01[cr] | 3.97 ± 0.13[bp] |
|  | 0.25 | 3.07 ± 0.04[aq] | 11.50 ± 0.11[ap] | 5.36 ± 0.09[bp] |
| 160 | 4 | 1.96 ± 0.01[bp] | 16.20 ± 0.02[bp] | 2.26 ± 0.02[cr] |
|  | 1 | 3.33 ± 0.05[ap] | 14.05 ± 0.05[bq] | 4.87 ± 0.12[cp] |
|  | 0.25 | 3.26 ± 0.12[ap] | 12.76 ± 0.11[ar] | 4.43 ± 0.11[ap] |

Different letters (a–c) in the same column indicate significant difference between maltodextrin concentrations ($p<0.05$). Different letters (p–r) in the same column indicate significant difference between inlet/outlet temperatures at $p<0.05$

mass [50] As surface adsorption and pore-filling mechanisms are governed by different heats of interactions, the difference in the time it takes for molecules to desorb from the sample bed serves as an indicator of the strength of their interaction. Furthermore, desorption isotherms can be acquired through either graphical or numerical integration methods [51]. In materials science and chemistry, BET surface area analysis is a crucial technique for assessing the surface properties of various materials.

Micropores are further classified into ultra-micropores ($d_{pore} < 0.7$ nm), medium-sized micropores (0.7 nm $< d_{pore} < 0.9$ nm), and super-micropores ($d_{pore} > 0.9$ nm) [52]. Nonporous materials and those primarily featuring macropores yield isotherms of type II (reversible isotherms), where $N_2$ molecules are absorbed in mono/multi-layers without restriction [29]. Experimental adsorption of nitrogen into intrinsically microporous material for extended durations can lead to irreversible polymer relaxations or swelling, accommodating the adsorbed nitrogen molecules and resulting in increased pore volume and, ultimately, increased adsorption. Beyond the micropore region (p/po $> 0.2$), the simulated isotherm reaches a plateau, indicating the saturation of all available micropores [53].

### 3.7 Powder quality attributes

The moisture content in the Noni powders varied between 1.96% and 4.87% (w/w). Table 8 illustrates the obtained values concerning inlet air temperature and the Noni juice-to-maltodextrin solids ratio for different maltodextrin dextrose equivalents. The greater the temperature disparity between the drying medium and the particles, the more rapid the heat transfer rate into the particles serving as the driving force for the removal of moisture [54].

The moisture content decreased as the maltodextrin concentration increased. A reduced moisture content contributes to improving the stability and shelf life of the product. Previous research indicated that the most minimal powder moisture content levels were attained with the highest concentration of maltodextrin. The powder's hygroscopicity denotes its ability to absorb moisture from the surrounding humid environment [55], and is generally considered desirable at lower levels along with lower moisture content and caking degree, and higher solubility for a high-quality powdered product. In the realm of powdered products, lower hygroscopicity is preferred as it helps prevent clumping and the formation of lumps, thereby preserving the product's texture and quality. When the inlet air temperature is excessively low

during the drying process, it may impede effective moisture removal, leading to a powder with increased hygroscopicity, making it more susceptible to moisture absorption from the environment.

As reported by Cai and Corke [34], when spray-drying cactus pear juice and betacyanin pigment, there was a decrease in hygroscopicity as the maltodextrin concentration increased [8]. Moisture is absorbed on the surfaces of particles, creating a saturated solution that renders the particles adhesive, allowing them to form liquid bridges. Consequently, the influence of process variables on the degree of powder caking is dependent on how they affect the hygroscopic nature of the powder [54]. Higher inlet air temperatures during the drying process often led to more efficient moisture removal, resulting in a powder with reduced moisture content and, consequently, a decreased degree of caking. Lower moisture content generally contributes to improved powder flowability and a decreased likelihood of clumping. Meanwhile, increasing the concentration of maltodextrin can act as a drying aid, absorbing excess moisture and forming a protective coating on the powder particles, contributing to a decrease in the degree of caking.

The aggregation of dehydrated powdered foods with high sugar content is attributed to the absorption of moisture, leading to the formation of a saturated solution on particle surfaces. This causes the particles to become sticky and facilitates the creation of liquid bridges, contributing to the formation of clumps.

Therefore, the influence of process variables on the extent of powder caking depends on how these variables affect the hygroscopic properties of the powder [54]. Elevated temperatures typically augment the vapor pressure gradient between the moist product and the drying air, facilitating a more rapid transfer of moisture. This is why the utilization of high-temperature air is frequently favored in industrial drying applications to expedite the drying process. Nevertheless, the specific conditions may vary depending on the attributes of the food product and the intended of product quality.

## 4. Conclusion

In conclusion, the changing food landscape, driven by our inclination for meals that resist diseases, requires ongoing exploration of inventive processing methods. Spray drying becomes a more practical option for various applications when applied to more adaptable equipment and smaller-scale experiments. The implementation of oxygen plasma treatment notably enhanced the hydrophobicity of the surface of the PTFE-coated plate. Plasma treatment, especially with oxygen plasma, has the capability to transform initially hydrophobic (water-repelling) surfaces into more hydrophilic (water-loving) ones. This proves advantageous in applications such as spray drying, where powders must be evenly dispersed and collected to achieve maximum yield. This enhanced functionality widens the range of applications for this versatile material. This is crucial for minimizing stickiness and wall deposition within the drying chamber of the spray dryer. The SEM results indicated that longer plasma treatment durations enhanced the contact angle by roughening the surface, revealing a correlation between power and treatment duration. RSM analysis highlighted the substantial influence of both power and treatment duration on the contact angle, with predicted values aligning well with experimental data. BET analysis showcased the sustained porosity of the PTFE-coated substrate after oxygen-plasma treatment, vital for subsequent drying cycles under the superhydrophobic regime. The surface morphology exhibited a porous and rough microstructural environment, with BET surface area analysis confirming an average particle size of less than 100 nm in all coated films. The developed technique for spray drying Noni juice using post-oxygenated PTFE substrate and maltodextrin as a drying agent demonstrated promising results. Lower moisture content,

reduced hygroscopicity, decreased caking, increased bulk density, and improved solubility were achieved under optimal conditions, leading to a high-quality powdered product. Surprisingly, maltodextrins positively influenced the drying process on the highly hydrophobic post-oxygenated PTFE substrate, facilitating faster drying conditions for Noni droplets and minimizing wall deposition. Surface modification via plasma treatment offers numerous benefits, making it a valuable asset across various industries. These advantages enhance product performance and functionality significantly. Unlike conventional solvent-based surface alteration techniques, plasma treatment provides a more environmentally sustainable alternative. It generates minimal waste and operates through a dry process. This discovery holds significance for industries such as food processing and pharmaceuticals, underscoring the importance of controlling the drying process to achieve superior product quality.

## Acknowledgments

The authors express their gratitude to Universiti Kebangsaan Malaysia (UKM) for the financial support extended to this project. Appreciation is also extended to the Department of Food Sciences and the Innovative Centre for Confectionery Technology (MANIS) within the Faculty of Science and Technology for their provision of research facilities and guidance.

## Author Contributions

**Conceptualization:** Saiful Irwan Zubairi, Mohamad Yusof Maskat.

**Investigation:** Noraziani Zainal Abidin.

**Methodology:** Noraziani Zainal Abidin, Noorain Purhanudin, Rozidawati Awang, Jarinah Mohd Ali, Harisun Yaakob.

**Project administration:** Haslaniza Hashim.

**Resources:** Haslaniza Hashim.

**Supervision:** Haslaniza Hashim, Saiful Irwan Zubairi.

**Validation:** Mohamad Yusof Maskat, Noorain Purhanudin, Rozidawati Awang, Jarinah Mohd Ali, Harisun Yaakob.

**Writing – original draft:** Noraziani Zainal Abidin.

**Writing – review & editing:** Haslaniza Hashim, Saiful Irwan Zubairi.

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
