## [Decision Letter · Decision Letter 0]

16 Apr 2024

PONE-D-24-05867Enhancing Polytetrafluoroethylene (PTFE) Coated Film for Food Processing: Unveiling Surface Transformations through Oxygenated Plasma Treatment and Parameter Optimization using Response Surface MethodologyPLOS ONE

Dear Dr. Zubairi,

Thank you for submitting your manuscript to PLOS ONE. After careful consideration, we feel that it has merit but does not fully meet PLOS ONE’s publication criteria as it currently stands. Therefore, we invite you to submit a revised version of the manuscript that addresses the points raised during the review process.

Reviewer 1:

Abstract: The excess description in the last three sentences of the abstract needs to be minimized and should be more conclusion-oriented.

Introduction: The introduction looks like a literature review. Could you please compare the areas of advancement of your current studies with the relevant articles cited in each paragraph?

Materials and Methods: Well written.

Result and Discussion: Well written.

Conclusion: Improve the language and quality of the keynotes in this section for the readers. Rewrite the closing statements.

Reviewer 2:

The paper focuses on a topical issue to reduce difficulties due to inherent stickiness arising due to the existence of sugars and organic acids with low molecular weight and a low glass transition temperature. This study expands the knowledge base on the application of oxygen plasma treatment on PTFE coated substrates to improve the efficiency of the spray drying process in order to provide a cost-effective solution for the food industry.

The purpose of the research is clearly stated, the objectives are well established and ambitious, the literature review has been critically followed in line with the state of the art.

The study is well-designed and well-written, the methods are suitable and the obtained results are well aligned with the purpose of this work. The research methodology is adequate, well described and sufficiently detailed.

The results are well illustrated in 7 Figures and 8 Tables. Statistical data processing is well performed.

The results are clearly presented. Discussions generally provide relevant explanations for obtained results. Some improvements need to be made in the discussions to better highlight the added value of the research.

Conclusions are drawn accordingly based on the data presented and highlight well the results of this study.

The references are relevant to this research topic.

Some aspect can be improved:

The innovation of this study is not well highlighted.

A more description is needed to better reveal the added value of this work.

What are the advantages, disadvantages and limitations of this approach?

The study has high application potential, but also contributes significantly to the development of the knowledge base in the field. Since this approach is of high quality and has great potential for both science and applications, I recommend its publication after a minor review.

We look forward to receiving your revised manuscript.

Kind regards,

Sara Hemati

Academic Editor

PLOS ONE

Journal Requirements:

"The authors express their gratitude to Universiti Kebangsaan Malaysia (UKM) for the financial support extended to this project (GUP-2020-080; ST-2023-043). Appreciation is also extended to the Department of Food Sciences and the Innovative Centre for Confectionery Technology (MANIS) within the Faculty of Science and Technology for their provision of research facilities and guidance."

"The authors express their gratitude to Universiti Kebangsaan Malaysia (UKM) for the financial support extended to this project (GUP-2020-080; ST-2023-043). Appreciation is also extended to the Department of Food Sciences and the Innovative Centre for Confectionery Technology (MANIS) within the Faculty of Science and Technology for their provision of research facilities and guidance."

"The authors express their gratitude to Universiti Kebangsaan Malaysia (UKM) for the financial support extended to this project (GUP-2020-080; ST-2023-043). Appreciation is also extended to the Department of Food Sciences and the Innovative Centre for Confectionery Technology (MANIS) within the Faculty of Science and Technology for their provision of research facilities and guidance."

5. We note that your Data Availability Statement is currently as follows: [All relevant data are within the manuscript and its Supporting Information files.]

Reviewers' comments:

Reviewer's Responses to Questions

**Comments to the Author**

1. Is the manuscript technically sound, and do the data support the conclusions?

Reviewer #1: Yes

Reviewer #2: Yes

2. Has the statistical analysis been performed appropriately and rigorously? 

Reviewer #1: Yes

Reviewer #2: Yes

3. Have the authors made all data underlying the findings in their manuscript fully available?

Reviewer #1: Yes

Reviewer #2: Yes

4. Is the manuscript presented in an intelligible fashion and written in standard English?

Reviewer #1: Yes

Reviewer #2: Yes

5. Review Comments to the Author

Reviewer #1: The manuscript entitled “Enhancing Polytetrafluoroethylene (PTFE) Coated Film for Food Processing: Unveiling Surface Transformations through Oxygenated Plasma Treatment and Parameter

Optimization using Response Surface Methodology” is worth publishing in PLOS ONE after some corrections.

Abstract: The excess description in the last three sentences of the abstract needs to be minimized and should be more conclusion-oriented.

Introduction: The introduction looks like a literature review. Could you please compare the areas of advancement of your current studies with the relevant articles cited in each paragraph?

Materials and Methods: Well written.

Result and Discussion: Well written.

Conclusion: Improve the language and quality of the keynotes in this section for the readers. Rewrite the closing statements.

Reviewer #2: The paper focuses on a topical issue to reduce difficulties due to inherent stickiness arising due to the existence of sugars and organic acids with low molecular weight and a low glass transition temperature. This study expands the knowledge base on the application of oxygen plasma treatment on PTFE coated substrates to improve the efficiency of the spray drying process in order to provide a cost-effective solution for the food industry.

The purpose of the research is clearly stated, the objectives are well established and ambitious, the literature review has been critically followed in line with the state of the art.

The study is well-designed and well-written, the methods are suitable and the obtained results are well aligned with the purpose of this work. The research methodology is adequate, well described and sufficiently detailed.

The results are well illustrated in 7 Figures and 8 Tables. Statistical data processing is well performed.

The results are clearly presented. Discussions generally provide relevant explanations for obtained results. Some improvements need to be made in the discussions to better highlight the added value of the research.

Conclusions are drawn accordingly based on the data presented and highlight well the results of this study.

The references are relevant to this research topic.

Some aspect can be improved:

The innovation of this study is not well highlighted.

A more description is needed to better reveal the added value of this work.

What are the advantages, disadvantages and limitations of this approach?

The study has high application potential, but also contributes significantly to the development of the knowledge base in the field. Since this approach is of high quality and has great potential for both science and applications, I recommend its publication after a minor review.

6. PLOS authors have the option to publish the peer review history of their article (what does this mean?). If published, this will include your full peer review and any attached files.

Reviewer #1: No

Reviewer #2: No

---

## [Author Response · Author response to Decision Letter 0]

30 Apr 2024

Dear Editor-in-Chief,

PLOS ONE

Thank you for the prompt review of our manuscript (Ref. No.: PONE-D-24-05867) entitled Enhancing Polytetrafluoroethylene (PTFE) Coated Film for Food Processing: Unveiling Surface Transformations through Oxygenated Plasma Treatment and Parameter Optimization using Response Surface Methodology. We are obliged to revise the manuscript as required and to resubmit it for publication in your PLOS ONE. We acted accordingly to improve the manuscript based on the comments of reviewers and resubmit the corrected manuscript as attached. Attached herewith (in the file submission system) are the lists of corrections for your kind perusal. 

Best regards, 

Assoc. Prof. Ts. Dr. SAIFUL IRWAN ZUBAIRI 

Department of Food Sciences, Faculty of Science & Technology, 

Universiti Kebangsaan Malaysia, 43600 UKM Bangi, Selangor.

+603-8921-5989 | saiful.zubairi08@alumni.imperial.ac.uk | saiful-z@ukm.edu.my

---

## [Editor Report · Decision Letter 1]

3 May 2024

Enhancing Polytetrafluoroethylene (PTFE) Coated Film for Food Processing: Unveiling Surface Transformations through Oxygenated Plasma Treatment and Parameter Optimization using Response Surface Methodology

PONE-D-24-05867R1

Dear Dr. Zainal Abidin,

We’re pleased to inform you that your manuscript has been judged scientifically suitable for publication and will be formally accepted for publication once it meets all outstanding technical requirements.

Kind regards,

Sara Hemati

Academic Editor

PLOS ONE

Additional Editor Comments (optional):

-

Reviewers' comments:

Accept

---

## [Editor Report · Acceptance letter]

8 May 2024

PONE-D-24-05867R1 

PLOS ONE

Dear Dr. Zubairi, 

I'm pleased to inform you that your manuscript has been deemed suitable for publication in PLOS ONE. Congratulations! Your manuscript is now being handed over to our production team.

Kind regards, 

on behalf of

Dr. Sara Hemati 

Academic Editor

PLOS ONE